# Berberine ameliorates vascular dysfunction by a global modulation of lncRNA and mRNA expression profiles in hypertensive mouse aortae

Na Tan[1], Yi Zhang[1], Yan Zhang[1,2], Li Li[1,2], Yi Zong[1], Wenwen Han[1], Limei Liu[1,2]*

1 Department of Physiology and Pathophysiology, School of Basic Medical Sciences, Peking University, Beijing, China, 2 Key Laboratory of Molecular Cardiovascular Science, Ministry of Education, Beijing, China

* liu_limei@126.com

## Abstract

### Objective

The current study investigated the mechanism underlying the therapeutic effects of berberine in the vasculature in hypertension.

### Methods

Angiotensin II (Ang II)-loaded osmotic pumps were implanted in C57BL/6J mice with or without berberine administration. Mouse aortae were suspended in myograph for force measurement. Microarray technology were performed to analyze expression profiles of lncRNAs and mRNAs in the aortae. These dysregulated expressions were then validated by qRT-PCR. LncRNA-mRNA co-expression network was constructed to reveal the specific relationships.

### Results

Ang II resulted in a significant increase in the blood pressure of mice, which was suppressed by berberine. The impaired endothelium-dependent aortic relaxation was restored in hypertensive mice. Microarray data revealed that 578 lncRNAs and 554 mRNAs were up-regulated, while 320 lncRNAs and 377 mRNAs were down-regulated in the aortae by Ang II; both were reversed by berberine treatment. qRT-PCR validation results of differentially expressed genes (14 lncRNAs and 6 mRNAs) were completely consistent with the microarray data. GO analysis showed that these verified differentially expressed genes were significantly enriched in terms of "cellular process", "biological regulation" and "regulation of biological process", whilst KEGG analysis identified vascular function-related pathways including cAMP signaling pathway, cGMP-PKG signaling pathway, and calcium signaling pathway etc. Importantly, we observed that lncRNA ENSMUST00000144849, ENSMUST00000155383, and AK041185 were majorly expressed in endothelial cells.

**Data Availability Statement:** The microarray data have been uploaded to NCBI Gene Expression Omnibus (GEO) under the accession number

GSE159725 (https://www.ncbi.nlm.nih.gov/geo/query/acc.cgi?acc=GSE159725).

**Funding:** This work was supported by the National Natural Science Foundation of China [81873475 (L. M. L), 81873719(Y. Z.), 81770225 (L. L.), 81570445(L. M. L)]. https://isisn.nsfc.gov.cn/egrantindex/funcindex/prjsearch-list. The funders had no role in study design, data collection and analysis, decision to publish, or preparation of the manuscript.

**Competing interests:** The authors have declared that no competing interests exist.

## Conclusion

The present results suggest that the five lncRNAs ENSMUST00000144849, NR_028422, ENSMUST00000155383, AK041185, and uc.335+ might serve critical regulatory roles in hypertensive vasculature by targeting pivotal mRNAs and subsequently affecting vascular function-related pathways. Moreover, these lncRNAs were modulated by berberine, therefore providing the novel potential therapeutic targets of berberine in hypertension. Furthermore, lncRNA ENSMUST00000144849, ENSMUST00000155383, and AK041185 might be involved in the preservation of vascular endothelial cell function.

## Introduction

Hypertension is a leading common risk factor for worldwide morbidity and mortality from cardiovascular diseases [1–3], which is associated with the alterations in the function and structure of resistance and conduit arteries. Endothelial dysfunction (ED) may be either a cause or a consequence of hypertension. ED contributes to increased systemic vascular resistance and thus leads to the development and maintenance of hypertension [4]; persistent hypertension conversely impairs vascular endothelial function through reducing endothelial nitric oxide synthase (eNOS) activity or/and enhancing endoplasmic reticulum (ER) stress and oxidative stress [5–7].

Berberine, an alkaloid extract from many Chinese medicinal herbs, has long been used to treat gastrointestinal infections and diarrheas. In recent decades, multiple beneficial influences of berberine on cardiovascular system have been reported, including anti-hyperglycemic, anti-oxidative, cholesterol-lowering, and cardiac protective effects [8–10]. Both animal studies and clinical experiments indicate that berberine exerts protective effects on vascular endothelial function. Berberine attenuates carotid arterial endothelium-dependent contractions by inhibiting ER stress [11] and ameliorates aortic endothelial dysfunction [12] in spontaneously hypertensive rats (SHR), thus postponing the progress of hypertension. It improves flow-mediated vasodilation (FMD) possibly via reducing circulating endothelial microparticles in human [13]. Combination therapy with berberine and trimetazidine significantly increases FMD in patients with coronary heart disease and primary hypertension [14].

Long noncoding RNAs (lncRNAs), a class of noncoding RNAs, are endogenously transcripts with length of > 200 nucleotides [15]. Emerging evidence point out aspects on the relevance of lncRNAs to endothelial dysfunction in hypertension. LncRNA-GAS5 regulates endothelial cell function through β-catenin signaling in hypertension [16]. LncRNA-FENDRR mediates vascular endothelial growth factor to promote the apoptosis of brain microvascular endothelial cells in the intracerebral hemorrhage of hypertensive mice [17]. LncRNA-AK094457 accelerates angiotensin II (Ang II)-induced aortic endothelial dysfunction in SHR [18]. Berberine has been shown to play protective effects via regulating lncRNAs in colorectal cancer [19], myocardial hypertrophy [20], and ischemic brain [21]. However, whether berberine could affect the expressions of lncRNAs in endothelial cells in hypertension remains still unknown.

In the present study, we evaluated the benefit of berberine on aortic endothelium-dependent relaxation in Ang II-mediated hypertensive mice. Furthermore, we performed systematical analyses to explore the regulatory effects of berberine on the lncRNA expressions in hypertensive mouse aortae and reveal the underlying molecular mechanisms. This study aimed at providing novel insights into the therapeutic role of berberine in hypertension and related cardiovascular diseases.

## Materials and methods

### Animal protocols

Male C57BL/6J mice (8–10 weeks old) were supplied by the Laboratory Animal Science Department of Peking University Health Science Center. Animal care and experimental procedures in this study were approved by the Animal Experimentation Ethics Committee of Peking University Health Science Center (LA2020122) and conformed to the Guide for the Care and Use of Laboratory Animals published by the US National Institute of Health (NIH Publication, 8th Edition, 2011).

Under ketamine/xylazine anesthesia (75 and 6 mg/kg), Ang II (1mg/kg/day)- or PBS-loaded osmotic pumps were implanted in mice. Then the mice received berberine administration (100 mg/kg/day) or vehicle in drinking water for 2 weeks. Systolic blood pressure (SBP) was measured by the tail-cuff method before and after treatment for two weeks. Ang II and berberine chloride were respectively purchased from Tocris Bioscience (Bristol, UK) and Sigma-Aldrich Chemical (St Louis, MO, USA). Ang II was dissolved in PBS. Berberine was dissolved in drinking water.

### Artery preparation and functional assay

Mice were sacrificed by $CO_2$ suffocation. Aortae from mice were removed and placed in ice-cold Krebs solution (mmol/L): 119 NaCl, 4.7 KCl, 1 $MgCl_2$, 2.5 $CaCl_2$, 1.2 $KH_2PO_4$, 25 $NaHCO_3$, and 11 D-glucose. Then aortae were cleaned of adhering tissue and cut into segments of ~1.8 mm in length. Changes in isometric tension of aortic rings were recorded in myograph (Danish Myo Technology, Aarhus, Denmark). Each ring was stretched to 3 mN, stabilized for 90 min, and then contracted with 60 mmol/L KCl. After several washes by warmed Krebs solution, endothelium-dependent relaxations (EDRs) in response to acetylcholine (ACh, 0.003 to 10 μmol/L) were examined in the aortae pre-contracted with phenylephrine (Phe, 1 μmol/L). Followed by washes, the rings were subjected to 30-minute exposure to $N^G$-nitro-$_L$-arginine methyl ester ($_L$-NAME, 100 μmol/L) and then contracted with Phe (1 μmol/L). Subsequently, endothelium-independent relaxations to cumulative additions of sodium nitroprusside (SNP, 0.001 to 10 μmol/L) were measured. Phe, ACh, $_L$-NAME, and SNP were purchased from Sigma-Aldrich Chemical (St Louis, MO, USA) and dissolved in distilled water.

### Total RNA extraction

After mice were sacrificed, aortae were removed, cleaned of adhering tissue in sterilized PBS, and then stored in liquid nitrogen for total RNA extraction. Total RNA from each sample was extracted using Trizol reagent (Invitrogen Life Technologies, Carlsbad, CA, USA). RNA quantity and quality were measured by NanoDrop ND-1000 to make sure the high purity of the isolated RNA, as indicated by A260/280 $\geq$1.90 before microarray and quantitative real-time polymerase chain reaction (qRT-PCR) experiments.

### Microarray analysis

Microarray analysis was performed by KangChen Bio-tech (Shanghai, China). Briefly, extracted RNA samples was first purified and amplified. Then, the total RNA was transcribed into fluorescent cRNA along the entire length of the transcripts without 3′ bias utilizing a random priming method (Arraystar Flash RNA Labeling Kit, Arraystar). Finally, the labeled cRNA was hybridized onto the Arraystar Mouse LncRNA Array v3.0. After washing, the arrays were scanned on an Agilent Scanner G2505C. Agilent Feature Extraction software (version

11.0.1.1) was used to analyze the acquired array images. Quantile normalization and subsequent data processing were performed with the Gene Spring GX v12.1 software package (Agilent Technologies). Quantile-normalized lncRNAs and mRNAs with "Present" or "Marginal" ("All Targets Value") flags were chosen for further data analysis. Differentially expressed lncRNAs and mRNAs were defined as fold change (the absolute ratio [no log scale] of normalized intensities between every two groups) >2.0 and p-value <0.05. The microarray data have been uploaded to NCBI Gene Expression Omnibus (GEO) under the accession number GSE159725 (*https://www.ncbi.nlm.nih.gov/geo/query/acc.cgi?acc=GSE159725*).

### Quantitative real-time PCR validation

qRT-PCR was performed using SYBR® High-Sensitivity qPCR SuperMix (NovoStart®, Shanghai, China) according to the manufacturer's instruction. All experiments were conducted at least three times. The expression level of each gene was determined by the $2^{-\Delta\Delta Ct}$ method. The mRNA levels were normalized by GAPDH. All primers were shown in S1 Table (lncRNAs) and S2 Table (mRNAs).

### Functional annotation of differentially expressed lncRNAs and mRNAs

Gene ontology (GO) term enrichment analysis was used to elucidate the biological significance of the differentially expressed genes (DEGs), including cellular component, molecular function and biological process. Kyoto Encyclopedia of genes and genomes (KEGG) pathway analysis was performed to identify pathways in which DEGs significantly enriched. Differentially regulated mRNAs were uploaded into the Database for Annotation, Visualization and Integrated Discovery (DAVID, *https://david.ncifcrf.gov/*) for annotation and functional analysis, including gene set enrichment analysis and mapping gene sets to the KEGG pathway [22]. The significant GO terms and pathways were determined using the Fisher's exact test, and false discovery rate (FDR) was utilized to correct the p-values. p-value < 0.05 denotes statistical significance.

### lncRNA-mRNA co-expression network

A coding-non-coding gene co-expression network (CNC network) was constructed based on the correlation analysis between the differentially expressed lncRNAs and mRNAs [23]. LncRNAs and mRNAs with Pearson correlation coefficients not less than 0.9, p-value≤ 0.05, and FDR≤ 1 were selected to draw the network by the program of cytoscape. According to the size of the enrichment factor, the top 10 terms or pathways were extracted.

### Statistical analysis

All data were presented as mean ± SEM. One-way ANOVA followed by Tukey's test and LSD for multiple comparisons were used in statistical analysis, which was performed in SPSS version 24.0 software. p-value < 0.05 was considered to be statistically significant.

A supplemental methods section can be found in the S1 File.

## Results

### Berberine lowers blood pressure and ameliorates aortic endothelial dysfunction in hypertensive mice

To confirm the therapeutic effect of berberine on hypertensive vasculature, we induced hypertension in mice with Ang II. As shown in Fig 1A, 2-week infusion with Ang II induced the marked increase of systolic blood pressure (SBP) in mice and co-treatment with berberine

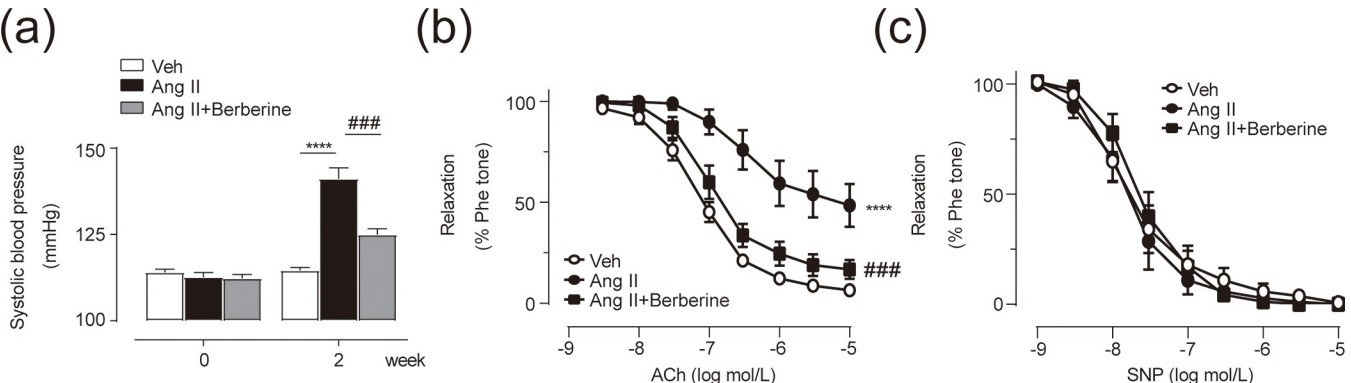

**Fig 1. Berberine lowered Systolic Blood Pressure (SBP) and improved Endothelium-Dependent Relaxation (EDR) in the aortae of Ang II-infused mice.**
(a) SBPs from Vehicle-, Ang II- and Ang II+Berberine-treated mice. (b) EDRs in the aortae from mice. (c) Endothelium-independent relaxations to sodium nitroprusside (SNP) in the mouse aortae. $^*p < 0.05$, $^{**}p < 0.01$, $^{***}p < 0.001$, $^{****}p < 0.0001$ *vs*. Veh, and #p $< 0.05$, ##p $< 0.01$, ###p $< 0.001$ *vs*. Ang II. The bars indicate the standard error of the mean (n = 5 for SBP; n = 4 for EDRs; n = 4 for SNP-induced relaxations). Veh, vehicle; Ang II, angiotensin II.

significantly lowered the SBP in hypertensive mice. Ang II-infused mice exhibited the impaired EDRs to ACh in the aortae, which were restored by berberine administration (Fig 1B). By contrast, endothelium-independent relaxations to SNP were similar among all the groups (Fig 1C). These data suggest that berberine ameliorated endothelial dysfunction in the conduit vessels in Ang II-mediated hypertension, thus attenuating the progression of hypertension.

## Microarray analysis for gene expression profiles

To identify possible lncRNAs and mRNAs participated in the efficacy of berberine, we performed a microarray-based analysis of gene expression profiles in the three groups of mice. By comparison of Ang II-treated group and Vehicle-treated group, we identified 2298 differentially expressed (DE) lncRNAs (including 1211 up-regulated and 1087 down-regulated, Fig 2A) and 2186 DE-mRNAs (including 1312 up-regulated and 874 down-regulated, Fig 2B). The expression levels of lncRNAs and mRNAs were dysregulated through over-expression or under-expression. Among 1545 DE-lncRNAs, 575 lncRNAs were up-regulated and 970 lncRNAs were down-regulated in Ang II+Berberine-treated group compared with Ang II-treated group (Fig 2C). Among 1590 DE-mRNAs, 702 mRNAs were significantly increased whereas 888 mRNAs were remarkably decreased in Ang II+Berberine-treated group (Fig 2D). Distinguishable expression patterns of top 60 lncRNAs and mRNAs were presented as the heat maps of hierarchical clustering in S1 Fig. This part of the results implicates that berberine might improve vascular function through regulating expressions of lncRNAs and mRNAs in hypertension.

In order to further understand the regulatory effects of berberine on gene expression in hypertensive vessels, the Venn diagram was performed to overlap the genes. Among the overlapped genes, the 320 decreased lncRNAs and 377 down-regulated mRNAs by Ang II were rescued after berberine administration; while berberine suppressed the up-regulated expressions of 578 lncRNAs and 554 mRNAs in hypertensive mouse aortae (Fig 3A and 3B). In order to have insight into the expression pattern of these dysregulated lncRNAs, we first divided the overlapped lncRNAs into two categories: 1–1 and 1–2 (1–1, genes up-regulated by Ang II but down-regulated by co-treatment with berberine; 1–2, genes suppressed by Ang II while rescued by berberine). The general signatures of these lncRNAs, including length distribution, classification, and chromosomal distribution, were thus summarized. These lncRNAs mainly ranged from 400 to 1200 bp in length (Fig 3C). In addition, Fig 3D revealed that these dysregulated lncRNAs mainly fell into six categories including exon sense-overlapping, intergenic,

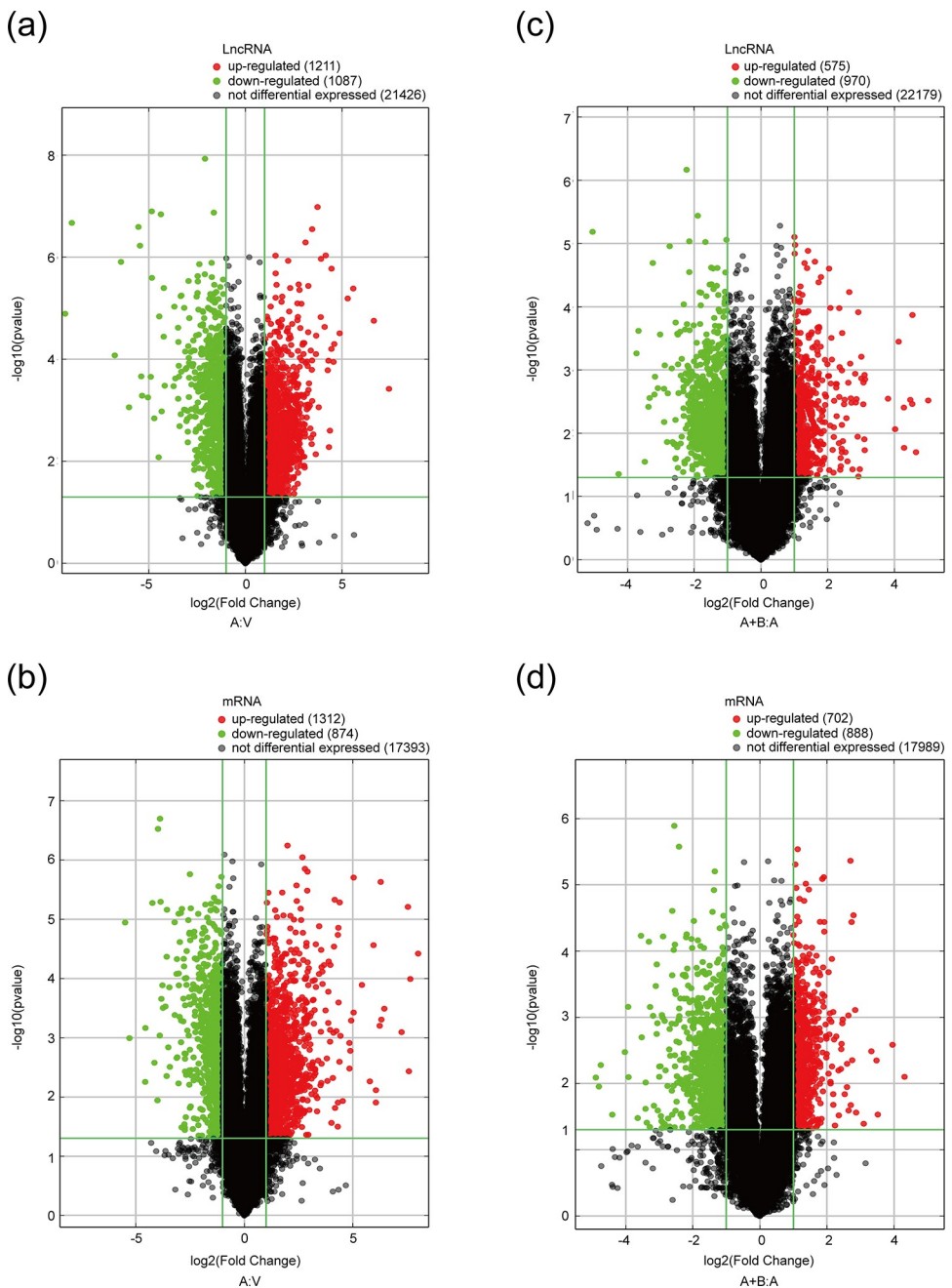

**Fig 2. Identification of the Differentially Expressed (DE) lncRNAs and mRNAs.** The Volcano plot analysis showed DE-lncRNAs in Ang II-treated group compared with Vehicle-treated group (a) and DE-lncRNAs in Ang II+Berberine co-treated group compared with Ang II-treated group (c); volcano plot identified the dysregulated mRNAs between Ang II and Vehicle groups (b) and Ang II+Berberine and Ang II groups (d). Red and green represented the up-regulated and down-regulated lncRNAs or mRNAs respectively; the vertical lines corresponded to 2-fold up-regulation or down-regulation, and the horizontal lines represented p = 0.05. A: V, Ang II (angiotensin II) *vs.* Vehicle. A+B: A, Ang II+Berberine *vs.* Ang II.

natural antisense, intronic antisense, bidirectional, and intron sense-overlapping. Moreover, more than 70% of the dysregulated lncRNAs were exon sense-overlapping and intergenic lncRNAs. In addition, we report that berberine-regulated lncRNAs were located at human chromosomes 1–19, X and Y. The under-expressed lncRNAs by berberine mostly located on 2

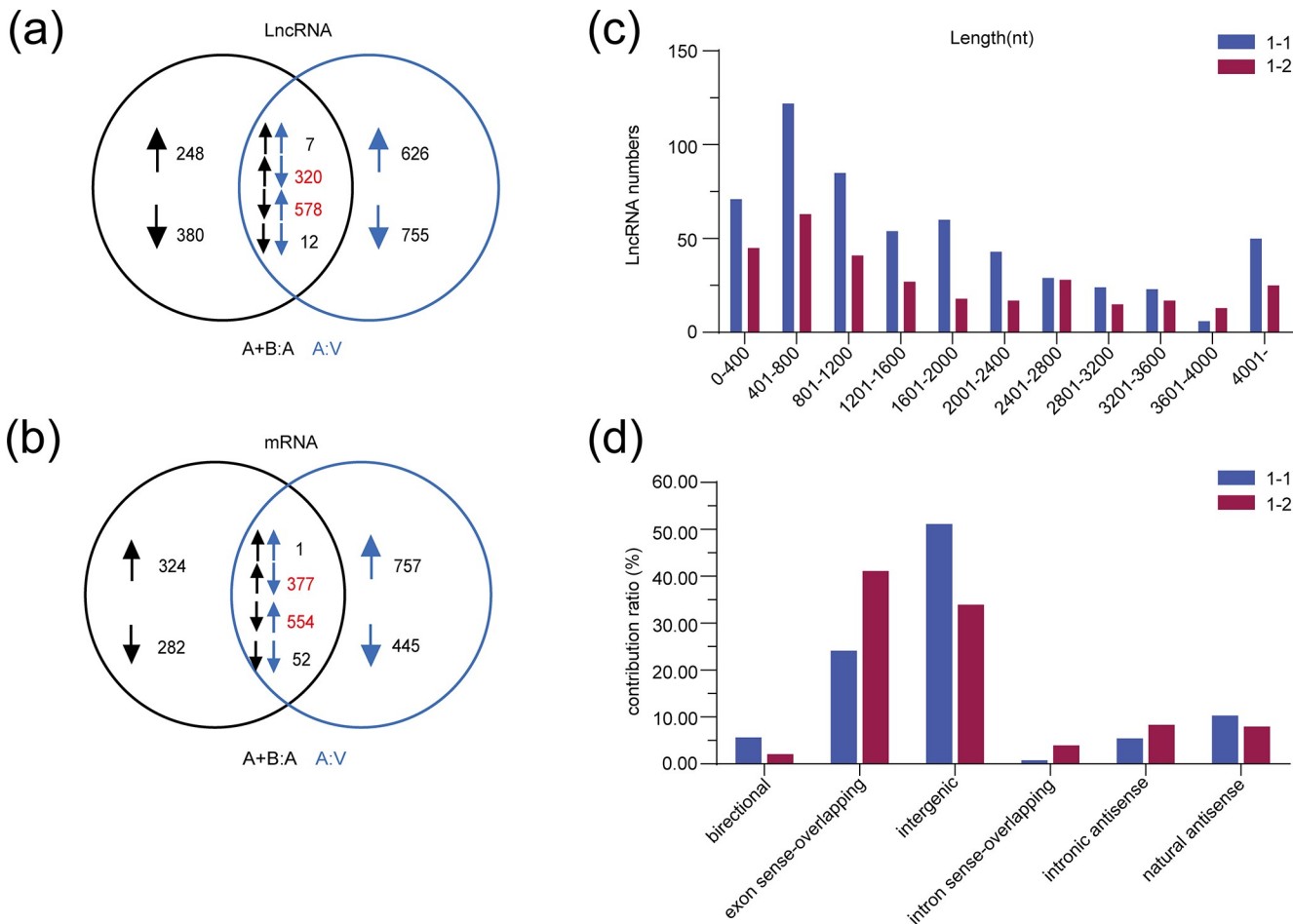

**Fig 3. The description of the microarray analysis.** Venn diagrams of differentially expressed (DE) lncRNAs (a) and DE-mRNAs (b) in the mouse aortae. The black or blue arrow represented the comparison of Ang II+Berberine and Ang II groups or Ang II and Vehicle groups. The up-arrows indicated up-regulation and the down-arrows indicated down-regulation. (c) The DE-lncRNAs were mainly between 400 and 1200 bp in length. (d) The relationship between these dysregulated lncRNAs and their targets. A: V, Ang II vs. Vehicle. A+B: A, Ang II+Berberine vs. Ang II; 1–1, genes up-regulated by Ang II but down-regulated by co-treatment with berberine; 1–2, genes suppressed by Ang II whilst reversed by berberine; Ang II, angiotensin II.

and 4 chromosomes (S2A Fig), whereas the elevated lncRNAs by berberine were mainly located at chromosome 2 and 11 (S2B Fig).

## GO analysis and KEGG pathway enrichment analysis

To analyze the functional enrichment and related signaling pathways of DEGs regulated by berberine in hypertension, the Gene Ontology (GO) along with the Kyoto Encyclopedia of Genes and Genomes (KEGG) were performed. The gene ontologies cover three domains including biological process (BP), cellular component (CC), and molecular function (MF). The top 10 significantly enriched GO terms in each domain were respectively shown in S3 and S4 Figs. In category 1–1 (S3 Fig), the meaningful BP terms are associated with immune system process (GO:0002376) and cell chemotaxis (GO:0060326); the DEGs of CC were obviously enriched in cell periphery (GO:0071944), plasma membrane (GO:0005886), and extracellular region (GO:0005576). In category 1–2 (S4 Fig), the DEGs were most significantly enriched in cellular process (GO:0009987) in the BP category; the DEGs under the CC were related to cell (GO:0005623) and cell part (GO:0044464). In addition, the terms in MF suggested that protein

binding (GO:0005515) and binding (GO:0005488) seemed particularly important in the improvement of vascular function by berberine (S3 and S4 Figs). KEGG pathway analysis revealed that Ang II might impair vascular function via up-regulating cytokine-cytokine receptor interaction, chemokine signaling, and cell adhesion molecules (S5A Fig) and suppressing cGMP-PKG signaling, vascular smooth muscle contraction, and ECM-receptor (S6A Fig). However, PPAR signaling, cytokine-cytokine receptor interaction, PI3K-Akt signaling, vascular smooth muscle contraction, and ECM-receptor were strongly influenced by berberine. The first three pathways (S5B Fig) were attenuated and the final two pathways (S6B Fig) were stimulated by berberine, thus improving vascular function in hypertension.

## qRT-PCR validation

To validate the dysregulated expressions of lncRNAs and mRNAs derived from microarray data, we performed Quantitative real-time polymerase chain reaction (qRT-PCR). 14 dysregulated lncRNAs, including 8 up-regulated DE-lncRNAs (Fig 4A: AK041185, AK044823,

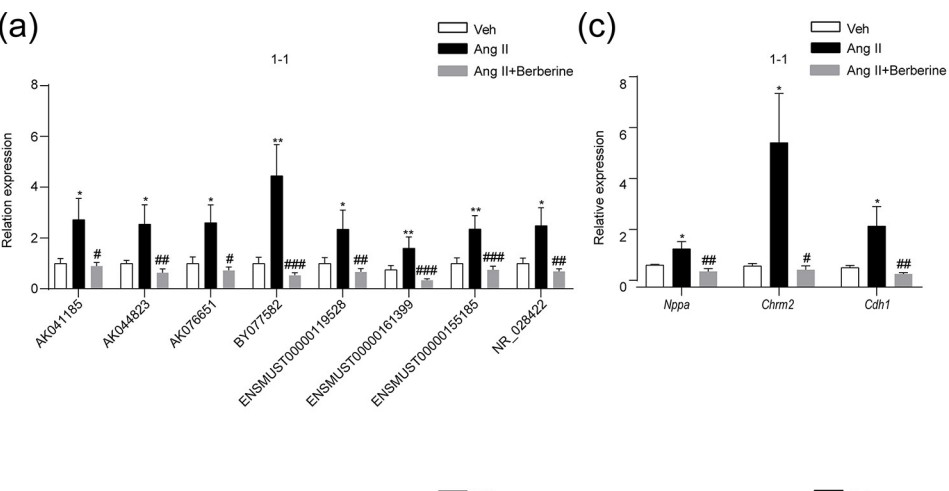

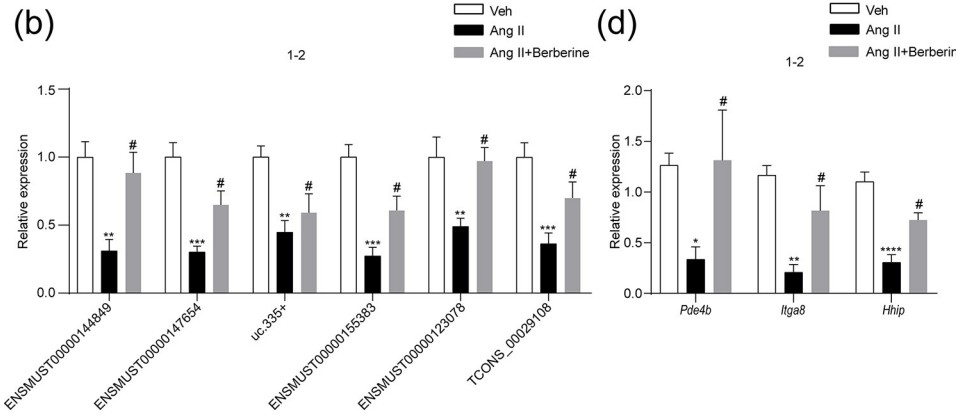

**Fig 4. Validation of DE-lncRNAs and mRNAs.** Quantitative RT-PCR demonstrated the dysregulated lncRNAs including 8 up-regulated DE-lncRNAs (a) and 6 down-regulated DE-lncRNAs (b) induced by Ang II in the aortae from mice; the dysregulated expressions of lncRNAs were modified by berberine administration (a, b). qRT-PCR validation results showed that 3 up-regulated (c) and 3 down-regulated (d) DE-mRNAs caused by Ang II were also modulated by berberine in hypertensive mouse aortae (c, d). $^*p < 0.05$, $^{**}p < 0.01$, $^{***}p < 0.001$, $^{****}p < 0.0001$ *vs.* Vehicle, and #p < 0.05, ##p < 0.01, ###p < 0.001 *vs.* Ang II. The bars indicate the standard error of the mean (n = 5). 1–1, genes up-regulated by Ang II but down-regulated by berberine; 1–2, genes suppressed by Ang II while increased by berberine; Veh, vehicle; Ang II, angiotensin II.

AK076651, BY077582, ENSMUST00000119528, ENSMUST00000161399, ENSMUST00000155185, and NR_028422) and 6 down-regulated DE-lncRNAs (Fig 4B: ENSMUST00000144849, ENSMUST00000147654, uc.335+, ENSMUST00000155383, ENSMUST00000123078, and TCONS_00029108) induced by Ang II, were successfully verified by qRT-PCR. Furthermore, these dysregulated expressions of lncRNAs were modified by berberine administration (Fig 4A and 4B). In the meanwhile, 3 up-regulated (*Nppa*, *Chrm2*, *and Cdh1*) and 3 down-regulated (*Pde4b*, *Itga8*, *Hhip*) DE-mRNAs by the induction of Ang II were selected for qRT-PCR validation. The validation results also revealed that berberine rescued the dysregulation of these DE-mRNAs in hypertensive vasculature (Fig 4C and 4D). The expression patterns of these lncRNAs and mRNAs in qRT-PCR validation results were consistent with those in microarray results. To sum up, these genes may play critical roles in the anti-hypertensive effect of berberine (the detailed information of the 14 lncRNAs and the 6 mRNAs was shown in the S3 and S4 Tables respectively).

### Construction of DE-lncRNA-mRNA co-expression network

To reveal the potential roles of hypertension-related lncRNAs and mRNAs, the DE-lncRNA-mRNA co-expression network was constructed. On the basis of lncRNA-mRNA co-expression network and genomic co-location, the roles of lncRNAs and mRNAs in the improvement of vascular function by berberine in hypertension were also predicted. GO enrichment indicated that the 14 DE-lncRNAs were involved in multiple molecular functions, such as "cellular process (GO:0009987)", "biological regulation (GO:0065007)" and "regulation of biological process (GO:0050789)" (Fig 5A). According to the KEGG pathway analysis, these DE-lncRNAs were enriched in the pathways of cell adhesion molecules, cAMP signaling pathway, cGMP-PKG signaling pathway, calcium signaling pathway, PI3K-Akt signaling pathway, vascular smooth muscle contraction, ECM-receptor interaction, adherens junction, and neuroactive ligand-receptor interaction, which were all associated with vascular function (Fig 5B). Based on the expression profiling, the significantly DE-mRNAs involved in these pathways were selected to construct the lncRNA-mRNA co-expression network; the latter included 14 aberrantly expressed lncRNAs and 137 most highly relevant dysregulated mRNAs including the 6 validated mRNAs (Fig 6).

### Discussion

Berberine and its derivatives have been shown to attenuate the development of hypertension by ameliorating vascular dysfunction [11, 12, 24]. The current study strengthens the evidence that berberine protects vascular endothelial function in hypertensive mice. However, the molecular mechanisms underlying the beneficial effects of berberine on hypertensive vasculature remained to be further explored. Therefore, we performed the microarray analysis to identify differentially expressed (DE) genes in the aortae from Ang II-induced hypertensive mice under the treatment with vehicle or berberine. We observed 2298 DE-lncRNAs and 2186 DE-mRNAs in hypertensive mice. Moreover, 1545 DE-lncRNAs and 1590 DE-mRNAs were identified in hypertensive mice after berberine treatment. We for the first time implicated that berberine might improve vascular function through regulating expressions of lncRNAs in hypertension.

LncRNAs, a novel class of non-coding RNAs, play critical roles at transcriptional and post-transcriptional levels in a variety of physiological and pathological processes [25]. In recent years, lncRNAs are emerged as novel modulators in the development of hypertension [18, 26, 27]. LncRNA TUG1 promotes the proliferation and migration of vascular smooth muscle cells (VSMCs) by affecting miR-145-5p/FGF10 axis in hypertensive state [28]. Here, we revealed

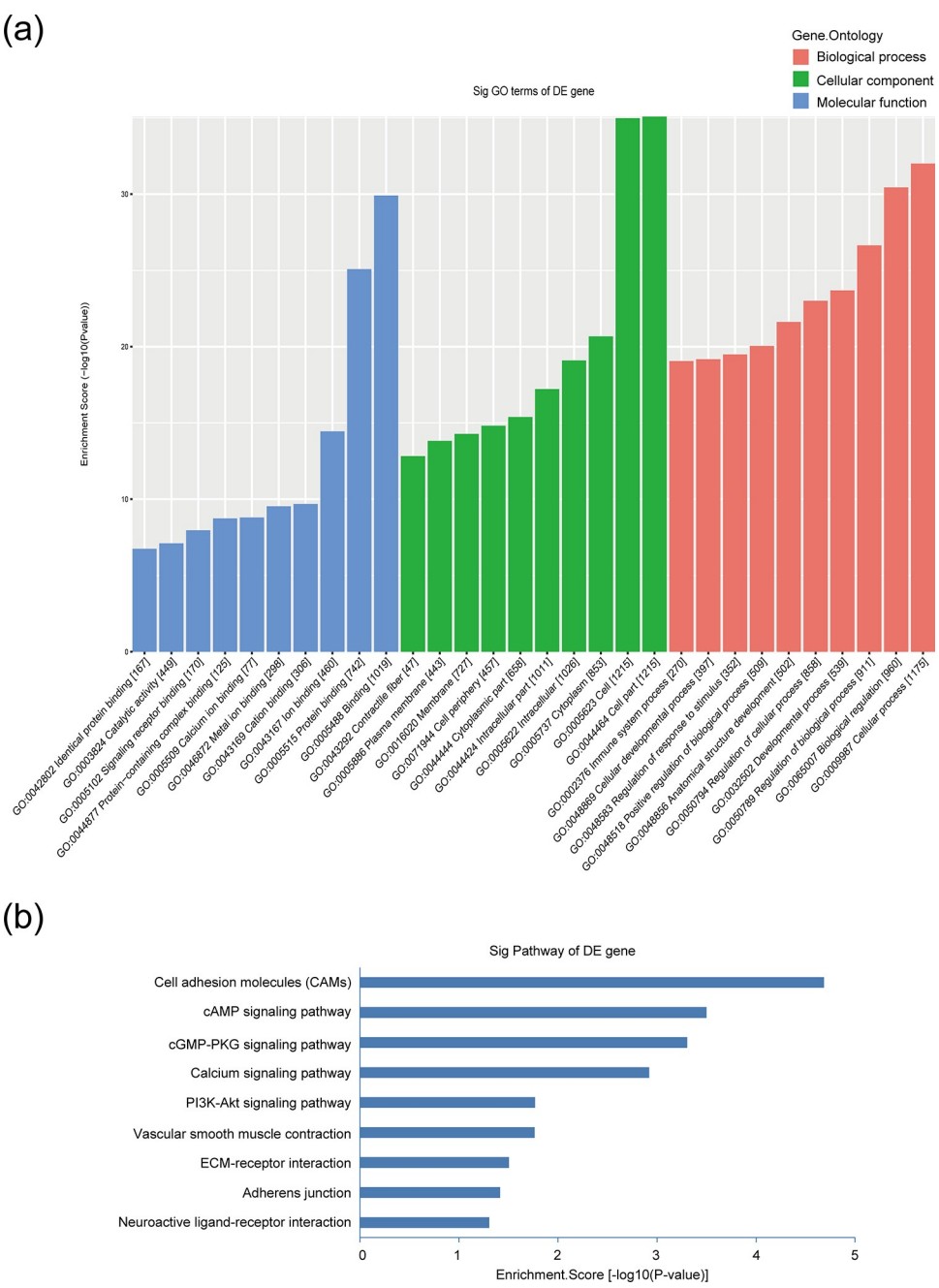

**Fig 5. GO and pathway analysis of the dysregulated lncRNAs-co-expressed mRNAs.** (a) The top 10 GO terms of biological process (blue), cellular components (green) and molecular functions (red) were listed. (b) The 9 pathways in which these 14 DE-lncRNAs were enriched.

that 1211 lncRNAs were up-regulated and 1087 were down-regulated among 2298 DE-lncRNAs in hypertensive mouse aortae, suggesting the vital roles of lncRNAs in the occurrence and development of vascular dysfunction in hypertension. Berberine has been reported to ameliorate nonalcoholic fatty liver disease by modulating hepatic lncRNA expression profiles and reverse down-regulated MRAK052686 [29]. Although a good deal of evidence suggests the relevance of lncRNAs to endothelial dysfunction in hypertension, it is still unclear whether

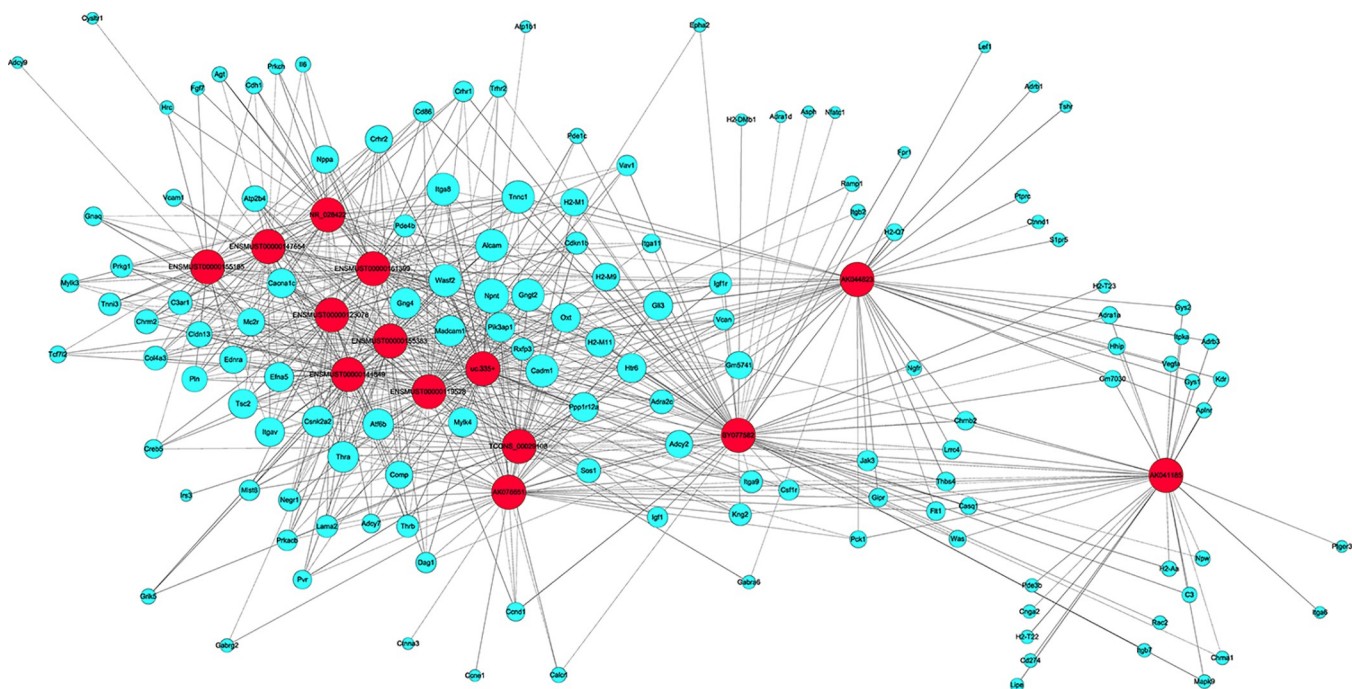

**Fig 6. The lncRNA-mRNA co-expression network.** It included 14 differentially expressed lncRNAs and 137 most highly relevant dysregulated mRNAs including the 6 mRNAs validated by qRT-PCR. Blue nodes were the coding genes; red nodes were lncRNAs. The solid line indicated a positive correlation and the dotted line indicated a negative correlation.

berberine improves vascular endothelial function by modulating lncRNA expressions in hypertension. Our microarray data observed 970 down-regulated and 575 up-regulated lncRNAs in hypertensive mice with the therapy of berberine.

Summing up the results of existing studies, we can learn that several proteins are involved in regulating the progression of high blood pressure, such as endothelin-1, Rac1, and G protein-coupled estrogen receptor [30–32], but hypertension-related genes have not been identified systemically. Here, we revealed that 2186 DE-mRNAs (including 1312 up-regulated mRNAs and 874 down-regulated mRNAs) were discovered in hypertensive mice. Simultaneously, 1590 DE-mRNAs were identified in berberine-treated hypertensive mice; among them were 702 elevated and 888 suppressed mRNAs. Next, the current study focused on the exploration of lncRNAs and mRNAs involved in the treatment with berberine to vascular function in hypertension. The GO analysis showed that 14 DE-lncRNAs (AK041185, AK044823, AK076651, BY077582, ENSMUST00000119528, ENSMUST00000161399, ENSMUST00000155185, NR_028422, ENSMUST00000144849, ENSMUST00000147654, uc.335+, ENSMUST00000155383, ENSMUST00000123078, and TCONS_00029108) were mainly involved in molecular functions, including biological regulation, cellular process, and regulation of biological process, which might be closely related to the regulation of vascular function by berberine. However, most of these lncRNAs were never reported both in the progress of hypertension and in the regulation of berberine on hypertensive vasculature. Importantly, we obtained the qRT-PCR validation results which were consistent with microarray results. This study comprehensively revealed the lncRNAs related to vascular dysfunction in hypertension. In addition, we provided a new perspective for the treatment of berberine on vascular dysfunction in hypertension.

Recent studies show that lncRNA-mRNA co-expression network analysis is widely used to evaluate the functional roles of lncRNAs, thus assessing the relationship between the lncRNA

expressions and the cardiovascular diseases. For example, Zhang *et al*. performed lncRNA-mRNA co-expression analysis in atherosclerosis [33] and Pant *et al*. constructed lncRNAs and mRNAs mediated co-expression network in diabetic cardiomyopathy [34]. Here, we also performed co-expression network analysis of the 14 DE-lncRNAs and the network consisted of 720 specific lncRNA-mRNA co-expression relationships. Interestingly, we observed these lncRNAs co-expressed with 137 sense mRNAs and were involved in many pathways, including cell adhesion molecules (CAMs), cAMP signaling pathway, cGMP-PKG signaling pathway, calcium signaling pathway, PI3K-Akt signaling pathway, vascular smooth muscle contraction, ECM-receptor interaction, adherens junction, and neuroactive ligand-receptor interaction, which were closely associated with vascular function. qRT-PCR validation also verified that the up-regulated (*Nppa*, *Chrm2*, and *Cdh1*) and the down-regulated (*Pde4b*, *Itga8*, *Hhip*) DE-mRNAs in the aortae from hypertensive mice were reversed by berberine; these data were consistent with the microarray data. Just as important, these 6 validated DE-mRNAs were included in the co-expression network. CAMs and ECM-receptor interaction signaling pathways are mainly involved in cell adhesion [35]. We found that the genes in the enrichments included the DE genes involved with the cadherin and integrin family (*Cdh1* and *Itga8*). According to our microassay results, *Cdh1* was co-expressed with lncRNA NR_028422 and *Itga8* was co-expressed with lncRNA ENSMUST00000155383 with extremely positively correlation; we also noticed that *Itga8* was also involved in PI3K-Akt signaling pathway. Except for natriuretic and diuretic effects, atrial natriuretic peptide (ANP) has been reported to elicit vasorelaxant action, thereby reducing body fluid volume and maintaining blood pressure homeostasis [36]. *Nppa* (encoding pro-ANP) has been shown to play a critical role in the reduction of SBP in hypertensive rats [37, 38]. Our current study announced that *Nppa* was elevated in hypertensive mouse aortae but attenuated by berberine, which might be closely related to lncRNA ENSMUST00000144849. Phosphodiesterases (PDE) limits the effects of vasodilators such as nitric oxide (NO). *Pde4b* hydrolyzes cyclic-AMP (cAMP); the latter suppresses PI3K/Akt signals and then restrains the vessel formation [39]. The expression of *Pde4b* was down-modulated in SHR preglomerular microvascular smooth muscle cells (SMCs) and endothelial cells (ECs) [40], but its expression can be induced by NO in rat pulmonary artery SMCs [41]. Our results also showed the reduced level of *Pde4b* in hypertension and the reversal effect of berberine on its expression. Furthermore, lncRNA uc.335+ might participate in regulating *Pde4b* in the therapy of berberine to hypertensive arterial function. *Chrm2*, associated with cAMP signaling pathway, has been reported to increase arterial contraction and lead to an increase of blood pressure [42]. We presented here an up-regulation of *Chrm2* and the inhibitory effect of berberine to *Chrm2* in the hypertensive mice. What's more, it was probably regulated by lncRNA NR_028422. *Hhip* gene, highly expressed in ECs, has a crucial role in controlling angiogenesis [43], yet the protective role of *Hhip* in the endothelial injury under hypertension is not understood. Both microarray analysis and qRT-PCR validation proved the under-expression of *Hhip* in hypertensive mouse aortae. Furthermore, the *Hhip* level was rescued and lncRNA AK041185 is possibly involved in the regulation of *Hhip* under the action of berberine. In order to explore whether the 5 lncRNAs were involved in the improvement of endothelial function in hypertension, we examined the expressions of the 5 lncRNAs both in ECs and in SMCs from C57BL/6J mouse aortae. As shown in S7 Fig, the levels of lncRNA ENSMUST00000144849, ENSMUST00000155383, and AK041185 were significantly higher in ECs than those in SMCs. LncRNA uc.335+ and lncRNA NR_028422 were also clearly expressed in ECs; but the expression level of lncRNA uc.335+ was similar in ECs to that in SMCs and the expression level of lncRNA NR_028422 in SMCs was obviously higher than that in ECs. Based on these observations, we speculated the potential benefits of ENSMUST00000155383, AK041185, and ENSMUST00000144849 in the maintenance of endothelial function.

## Conclusion

Taken together, these findings indicate that lncRNA ENSMUST00000144849, NR_028422, ENSMUST00000155383, AK041185, and uc.335+ may play critical roles by modulating the relevant mRNAs in the remedy of berberine to hypertensive vascular damage. This study provides novel insights of how lncRNAs affect hypertension and which pathways may play important roles during the condition. These results could help understand the biological mechanisms of berberine on the improvement of vascular function and structure in treating hypertension.

## Limitations

Our study revealed the possibility that lncRNA ENSMUST00000144849, ENSMUST00000155383, and AK041185 participated in vascular dysfunction in hypertension and could be regulated by berberine. Although we verified that they were majorly expressed in ECs, we did not make it clear whether the 3 lncRNAs were exactly involved in the improvement of vascular endothelial function in hypertension. Therefore, we should explore the expressions of these 3 lncRNAs in endothelial cells (ECs) from hypertensive mice and the regulatory effects of berberine on them in ECs under hypertensive state. In addition, the biology functions of lncRNA ENSMUST00000144849, ENSMUST00000155383, and AK041185 ought to be investigated by overexpression or knockdown of the lncRNAs, thus proving the correlation about the changes of the 3 lncRNAs and related mRNAs with the berberine treatment in hypertension. However, this is beyond the scope of the present study.

## Supporting information

**S1 Table. The primers of the lncRNAs.**
(DOCX)

**S2 Table. The primers of the mRNAs.**
(DOCX)

**S3 Table. Functional annotation of differentially expressed lncRNAs.**
(DOCX)

**S4 Table. Functional annotation of differentially expressed mRNAs.**
(DOCX)

**S1 File. Supplemental methods section.**
(DOCX)

**S1 Fig. Hierarchical clustering analysis.** It showed top 60 differentially expressed lncRNAs (a) and mRNAs (b) among the three groups. Red color indicates highly relative expression and green color indicates low relative expression. V, Vehicle; A, Ang II, angiotensin II; B, Ang II+-Berberine.
(TIF)

**S2 Fig.** Chromosomal distribution of the 1–1 (a) and 1–2 (b) lncRNAs. 1–1, genes up-regulated by Ang II but down-regulated by co-treatment with berberine; 1–2, genes suppressed by Ang II while reversed by berberine. Ang II, angiotensin II.
(TIF)

**S3 Fig. GO enrichment analysis of category 1–1.** Red, green and blue bars represented biological process (BP), cellular component (CC), and molecular function (MF). (a) The top 10

GO terms that were associated with the coding gene function of up-regulated lncRNAs in the Ang II-treated group compared with Vehicle-treated group. (b) the top 10 GO terms that were associated with the coding gene function of down-regulated lncRNAs in the Ang II+Berberine co-treated group compared with Ang II-treated group. 1–1, genes up-regulated by Ang II but down-regulated by berberine; V, Vehicle; A, Ang II, angiotensin II; A+B, Ang II+Berberine. (TIF)

**S4 Fig. GO enrichment analysis of category 1–2.** Red, green and blue bars represented biological process (BP), cellular component (CC), and molecular function (MF). (a) The top 10 GO terms that were associated with the coding gene function of down-regulated lncRNAs in the Ang II-treated group compared with Vehicle-treated group. (b) the top 10 GO terms that were associated with the coding gene function of up-regulated lncRNAs in the Ang II+-Berberine co-treated group compared with Ang II-treated group. 1–2, genes suppressed by Ang II while increased by berberine; V, Vehicle; A, Ang II, angiotensin II; A+B, Ang II+-Berberine. (TIF)

**S5 Fig. KEGG pathway analysis of 1–1.** It revealed that (a) the top 10 pathways that were associated with the coding gene function of up-regulated lncRNAs in the Ang II-treated group compared with Vehicle-treated group, (b) the top 10 pathways that were associated with the coding gene function of down-regulated lncRNAs in the Ang II+Berberine co-treated group compared with Ang II-treated group. 1–1, genes up-regulated by Ang II but down-regulated by berberine; V, Vehicle; A, Ang II, angiotensin II; A+B, Ang II+Berberine. (TIF)

**S6 Fig. KEGG pathway analysis of 1–2.** It showed (a) the top 10 pathways that were associated with the coding gene function of down-regulated lncRNAs in the Ang II-treated group compared with Vehicle-treated group, (b) the top 10 pathways that were associated with the coding gene function of up-regulated lncRNAs in the Ang II+Berberine co-treated group compared with Ang II-treated group. 1–2, genes suppressed by Ang II while increased by berberine; V, Vehicle; A, Ang II, angiotensin II; A+B, Ang II+Berberine. (PNG)

**S7 Fig. The expressions of the five lncRNAs in primary mouse aortic ECs and SMCs.** LncRNA ENSMUST00000144849, lncRNA ENSMUST00000155383, lncRNA AK041185, and lncRNA NR_028422 exhibited different expression levels in ECs and SMCs; but lncRNA uc.335+ expression was similar in the two cell types. EC, endothelial cells; SMCs, smooth muscle cells. (TIF)

## Author Contributions

**Conceptualization:** Na Tan, Limei Liu.

**Data curation:** Na Tan, Yi Zhang.

**Formal analysis:** Na Tan.

**Funding acquisition:** Limei Liu.

**Investigation:** Na Tan, Yi Zong, Wenwen Han.

**Methodology:** Na Tan.

**Project administration:** Na Tan.

**Resources:** Na Tan, Yi Zhang.

**Software:** Na Tan.

**Supervision:** Na Tan.

**Validation:** Na Tan, Yi Zhang, Limei Liu.

**Visualization:** Na Tan, Yan Zhang, Li Li, Limei Liu.

**Writing – original draft:** Na Tan.

**Writing – review & editing:** Limei Liu.

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
