## [Decision Letter · Decision Letter 0]

6 Jan 2021

PONE-D-20-35678

Berberine ameliorates vascular dysfunction by a global modulation of lncRNA and mRNA expression profiles in hypertensive mouse aortae

PLOS ONE

Dear Dr. Liu,

Thank you for submitting your manuscript to PLOS ONE. After careful consideration, we feel that it has merit but does not fully meet PLOS ONE’s publication criteria as it currently stands. Therefore, we invite you to submit a revised version of the manuscript that addresses the points raised during the review process.

The authors provided evidence for the potential mechanisms responsible for the beneficial effect of berberine against Ang II-induced hypertension and showed that in Ang II treatment increased the levels of 578 lncRNAs and 554 mRNAs while reduces the levels of 320 lncRNAs and 377 mRNAs in mouse aortas, and these changes can be reversed by berberine treatment in Ang II-infused mice. In general, this experiments were well designed and the results were clearly presented. However, both reviewers raised a number of constructive comments and suggestions for the authors to address.

They include

1. Is there any study showing the association between the 5 LncRNAs with blood pressure regulation?

2. Are the 5 LncRNAs majorly expressed in endothelial cells?

3. More information is necessary for the animal protocol (i) What is the dosage of AngII osmotic pumps implanted in the mice? (ii) What are sources and the solvents for Ang II and berberine? And (iii) Is berberine administered by oral gavage or other method?  

4. The authors shall briefly describe the limitations of the current study in the Discussion.

5. There are correlation about the changes of expression profiles of lnc RNA and mRNAs with the berberine treatment in hypertension. However, it is unclear whether such changes lead to the improvement of endothelial function. Future investigations are needed to confirm that the changes lead to endothelial dysfunction in hypertension in relation to the vaso-protective effect of berberine.

We look forward to receiving your revised manuscript.

Kind regards,

Yu Huang

Academic Editor

PLOS ONE

Journal Requirements:

2. Please ensure that in your methods section you have provided details of the sources of all materials, chemicals, equipment and instrumentation used in your study, including manufacturer/supplier names where relevant. This is in line with our reproducibility criterion for publishing, see https://journals.plos.org/plosone/s/criteria-for-publication#loc-3

Reviewers' comments:

Reviewer's Responses to Questions

**Comments to the Author**

1. Is the manuscript technically sound, and do the data support the conclusions?

Reviewer #1: Yes

Reviewer #2: Yes

2. Has the statistical analysis been performed appropriately and rigorously? 

Reviewer #1: Yes

Reviewer #2: Yes

3. Have the authors made all data underlying the findings in their manuscript fully available?

Reviewer #1: Yes

Reviewer #2: Yes

4. Is the manuscript presented in an intelligible fashion and written in standard English?

Reviewer #1: Yes

Reviewer #2: Yes

5. Review Comments to the Author

Reviewer #1: Na Tan et al performed the microarray and qRT-PCR to show the regulatory effects of berberine on the expression profiles of lnc RNA and mRNAs in hypertensive mouse aortae. The experiments provide significant and novel results about the molecular mechanisms of berberine in hypertension; and the manuscript are well written. I only have a few minor suggestions.

Sugesstions:

1. More information is necessary for the animal protocol:

i. What is the dosage of AngII osmotic pumps implanted in the mice?

ii. What are sources and the solvents for Ang II and berberine?

iii. Is berberine administered by oral gavage or other method?

2. The authors can mention the limitations of current study in the discussion part. There are correlation about the changes of expression profiles of lnc RNA and mRNAs with the berberine treatment in hypertension; nevertheless, it is not sure whether such changes lead to the improvement of endothelial function. Further investigations are needed to prove that the changes lead to the endothelial dysfunction in hypertension and vaso-protective effects of berberine.

Reviewer #2: This study mainly investigated the potential mechanism underlying the therapeutic effects of berberine against Ang II-induced hypertension. By using microarray technology, the authors revealed that in Ang II group, 578 lncRNAs and 554 mRNAs were up-regulated, 320 lncRNAs and 377 mRNAs were downregulated in the aortae, while some of them were reversed by berberine treatment. By deep analysis of these differentially expressed (DE) genes, they found that these DE genes were closely associated with NO production and vascular tone regulation. Based on these findings, the authors proposed 5 potential LncRNAs that may participate in the blood pressure-lowering effect of berberine. In general, this study is well designed and written. However, I have several concerns.

1. Is there any study showing the association between the 5 LncRNAs with blood pressure regulation?

2. Are the 5 LncRNAs majorly expressed in endothelial cells?

6. PLOS authors have the option to publish the peer review history of their article (what does this mean?). If published, this will include your full peer review and any attached files.

Reviewer #1: **Yes: **Wai San Cheang

Reviewer #2: **Yes: **Chenglin Zhang

---

## [Author Response · Author response to Decision Letter 0]

29 Jan 2021

The authors thank the constructive comments and suggestions of the two reviewers and we respond point-by-point as follows. We performed some new experiments to support our conclusion and also made necessary discussion.

Response to Reviewers' Comments:

Reviewer 1

Na Tan et al performed the microarray and qRT-PCR to show the regulatory effects of berberine on the expression profiles of lncRNA and mRNAs in hypertensive mouse aortae. The experiments provide significant and novel results about the molecular mechanisms of berberine in hypertension; and the manuscript are well written. I only have a few minor suggestions.

Authors’ response: We thank this reviewer for the very positive appraisal of our findings and manuscript.

1. More information is necessary for the animal protocol:

1) What is the dosage of Ang Ⅱ osmotic pumps implanted in the mice?

Authors’ response: The dosage of Ang Ⅱ is 1mg/kg/day. We revised with the detail in Materials and Methods section (please see page 4, line 106 in the revised manuscript) as “Ang II (1mg/kg/day)- or PBS-loaded osmotic pumps were implanted in mice”.

2) What are sources and the solvents for Ang Ⅱ and berberine?

Authors’ response: Ang Ⅱ and berberine chloride were respectively purchased from Tocris Bioscience (Bristol, UK) and Sigma-Aldrich Chemical (St Louis, MO, USA). We revised with the detail in Materials and Methods section (please see page 4, lines 110-112 in the revised manuscript) as “Ang II and berberine chloride were respectively purchased from Tocris Bioscience (Bristol, UK) and Sigma-Aldrich Chemical (St Louis, MO, USA). Ang II was dissolved in PBS. Berberine was dissolved in drinking water”.

3) Is berberine administered by oral gavage or other method?

Authors’ response: The mice received berberine administration (100 mg/kg/day) or vehicle in drinking water. We revised with the detail in Materials and Methods section (please see page 4, line 108 in the revised manuscript) as “Then the mice received berberine administration (100 mg/kg/day) or vehicle in drinking water for 2 weeks.”

2. The authors can mention the limitations of current study in the discussion part. There are correlation about the changes of expression profiles of lncRNA and mRNAs with the berberine treatment in hypertension; nevertheless, it is not sure whether such changes lead to the improvement of endothelial function. Further investigations are needed to prove that the changes lead to the endothelial dysfunction in hypertension and vaso-protective effects of berberine.

Authors’ response: We thank this reviewer for the constructive suggestion. We discussed the limitations in the revised manuscript (Page 12, lines 465-477).

Reviewer 2

This study mainly investigated the potential mechanism underlying the therapeutic effects of berberine against Ang II-induced hypertension. By using microarray technology, the authors revealed that in Ang II group, 578 lncRNAs and 554 mRNAs were up-regulated, 320 lncRNAs and 377 mRNAs were downregulated in the aortae, while some of them were reversed by berberine treatment. By deep analysis of these differentially expressed (DE) genes, they found that these DE genes were closely associated with NO production and vascular tone regulation. Based on these findings, the authors proposed 5 potential LncRNAs that may participate in the blood pressure-lowering effect of berberine. In general, this study is well designed and written. However, I have several concerns.

Authors’ response: We thank this reviewer for the positive appraisal of our findings and manuscript.

1. Is there any study showing the association between the 5 LncRNAs with blood pressure regulation?

Authors’ response: Till now, there is no evidence to state the association between the 5 LncRNAs with blood pressure regulation. But we will further explore the regulatory effects of these 5 lncRNAs on endothelial function and blood pressure in hypertension in future study. In addition, we discussed this limitation in the revised manuscript (Page 12, lines 465-477).

2. Are the 5 LncRNAs majorly expressed in endothelial cells?

Authors’ response: In order to explain this point, we primarily cultured endothelial cells (ECs) and smooth muscle cells (SMCs) of the aortae from C57/6J mice. We performed qRT-PCR to examine the expressions of the 5 lncRNAs both in ECs and in SMCs. As shown in S7 Fig, the levels of lncRNA ENSMUST00000155383, AK041185, and ENSMUST00000144849 were significantly higher in ECs than those in SMCs. LncRNA uc.335+ and lncRNA NR_028422 were also clearly expressed in ECs; but the expression level of lncRNA uc.335+ was similar in ECs and SMCs and the expression level of lncRNA NR_028422 in SMCs was higher than that in ECs. Moreover, we discussed this in the revised manuscript (Page 11, lines 446-455).

---

## [Decision Letter · Decision Letter 1]

10 Feb 2021

Berberine ameliorates vascular dysfunction by a global modulation of lncRNA and mRNA expression profiles in hypertensive mouse aortae

PONE-D-20-35678R1

Dear Dr. Liu

We’re pleased to inform you that your manuscript has been judged scientifically suitable for publication and will be formally accepted for publication once it meets all outstanding technical requirements.

Kind regards,

Yu Huang

Academic Editor

PLOS ONE

Additional Editor Comments (optional):

Reviewers' comments:

Reviewer's Responses to Questions

**Comments to the Author**

1. If the authors have adequately addressed your comments raised in a previous round of review and you feel that this manuscript is now acceptable for publication, you may indicate that here to bypass the “Comments to the Author” section, enter your conflict of interest statement in the “Confidential to Editor” section, and submit your "Accept" recommendation.

Reviewer #1: All comments have been addressed

Reviewer #2: All comments have been addressed

2. Is the manuscript technically sound, and do the data support the conclusions?

Reviewer #1: Yes

Reviewer #2: Yes

3. Has the statistical analysis been performed appropriately and rigorously? 

Reviewer #1: Yes

Reviewer #2: Yes

4. Have the authors made all data underlying the findings in their manuscript fully available?

Reviewer #1: Yes

Reviewer #2: Yes

5. Is the manuscript presented in an intelligible fashion and written in standard English?

Reviewer #1: Yes

Reviewer #2: Yes

6. Review Comments to the Author

Reviewer #1: (No Response)

Reviewer #2: (No Response)

7. PLOS authors have the option to publish the peer review history of their article (what does this mean?). If published, this will include your full peer review and any attached files.

Reviewer #1: **Yes: **Wai San Cheang

Reviewer #2: No

---

## [Editor Report · Acceptance letter]

15 Feb 2021

PONE-D-20-35678R1 

Berberine ameliorates vascular dysfunction by a global modulation of lncRNA and mRNA expression profiles in hypertensive mouse aortae 

Dear Dr. Liu:

I'm pleased to inform you that your manuscript has been deemed suitable for publication in PLOS ONE. Congratulations! Your manuscript is now with our production department. 

Kind regards, 

on behalf of

Dr. Yu Huang 

Academic Editor

PLOS ONE